# Predicting Hepatotoxicity Associated with Low-Dose Methotrexate Using Machine Learning

**DOI:** 10.3390/jcm12041599

**Published:** 2023-02-17

**Authors:** Qiaozhi Hu, Hualing Wang, Ting Xu

**Affiliations:** 1Department of Pharmacy, West China Hospital, Sichuan University, Chengdu 610041, China; 2West China School of Pharmacy, Sichuan University, Chengdu 610041, China

**Keywords:** low-dose methotrexate, hepatotoxicity, machine learning, prediction model, risk factor

## Abstract

An accurate prediction of the hepatotoxicity associated with low-dose methotrexate can provide evidence for a reasonable treatment choice. This study aimed to develop a machine learning-based prediction model to predict hepatotoxicity associated with low-dose methotrexate and explore the associated risk factors. Eligible patients with immune system disorders, who received low-dose methotrexate at West China Hospital between 1 January 2018, and 31 December 2019, were enrolled. A retrospective review of the included patients was conducted. Risk factors were selected from multiple patient characteristics, including demographics, admissions, and treatments. Eight algorithms, including eXtreme Gradient Boosting (XGBoost), AdaBoost, CatBoost, Gradient Boosting Decision Tree (GBDT), Light Gradient Boosting Machine (LightGBM), Tree-based Pipeline Optimization Tool (TPOT), Random Forest (RF), and Artificial Neural Network (ANN), were used to establish the prediction model. A total of 782 patients were included, and hepatotoxicity was detected in 35.68% (279/782) of the patients. The Random Forest model with the best predictive capacity was chosen to establish the prediction model (receiver operating characteristic curve 0.97, accuracy 64.33%, precision 50.00%, recall 32.14%, and F1 39.13%). Among the 15 risk factors, the highest score was a body mass index of 0.237, followed by age (0.198), the number of drugs (0.151), and the number of comorbidities (0.144). These factors demonstrated their importance in predicting hepatotoxicity associated with low-dose methotrexate. Using machine learning, this novel study established a predictive model for low-dose methotrexate-related hepatotoxicity. The model can improve medication safety in patients taking methotrexate in clinical practice.

## 1. Introduction

Methotrexate (MTX), a folic acid antagonist that inhibits dihydrofolate reductase in the S-phase cell cycle, was first developed as an anticancer treatment in the 1940s [1,2]. Since the 1950s, MTX has been prescribed as an immunosuppressant for treating immune system disorders, including rheumatoid arthritis, psoriasis, psoriatic arthritis, and inflammatory bowel diseases [3]. The overall prevalence of rheumatoid arthritis is 0.24% to 1.1% [4,5,6]. Similarly, a population-based study in the United States found that psoriasis rates increased significantly from 50.8 cases per 100,000 (from 1970 to 1974) to 100.5 cases per 100,000 (from 1995 to 1999) [7]. A worldwide review showed that the prevalence of psoriasis ranged from 0.5 to 11.4% in adults [8]. MTX is now recommended as a first- or second-line treatment for many immune system diseases [9,10,11,12,13,14,15]. Although several biological agents have emerged for immune system diseases in the last two decades, such as adalimumab [16], infliximab [17], canakinumab [18], ustekinumab [19], and secukinumab [19], MTX is still widely used due to its efficacy, low cost, and ease of administration. MTX can be administered orally or subcutaneously as a weekly treatment regimen [11].

Due to lower doses of MTX, life-threatening adverse drug effects (ADE) are rarely observed in MTX treatment for immune system diseases. However, severe ADEs can still occur, especially hepatotoxicity [20]. Abnormal serum levels of alanine aminotransferase (ALT) or aspartate aminotransferase (AST) occurred in 23.47% (315/1342) of patients treated for MTX-treated rheumatoid arthritis [21]. Liver enzyme abnormalities are the leading cause of dose modification or discontinuation of MTX [22]. Furthermore, a systematic review indicated that 33% of patients with psoriasis who received low-dose MTX had liver disease progression, such as liver fibrosis [23]. Therefore, it is crucial to clarify the incidence of liver ADE in MTX-treated patients. Risk factors can affect MTX therapies. These risk factors include alcohol use, history of liver disease, obesity, type 2 diabetes, history of significant exposure to hepatotoxic drugs or chemicals, lack of folate supplementation, and hyperlipidemia [24,25]. However, current research lacks an assessment of the impact of these risk factors. Furthermore, there is a lack of research exploring unknown risk factors and establishing predictive models for hepatotoxicity associated with low-dose MTX.

Machine learning is one of the fastest-growing technical fields [26] and has been widely used in medical fields, such as medical diagnosis and prediction of disease risks [27,28,29]. Machine learning can promote data-driven estimation when selecting multiple variables and processing complex nonlinear relationships among multidimensional variables [30]. Therefore, machine learning can increase the precision of prediction models, especially for analyzing large datasets with many variables [31,32]. The study aimed to compare eight machine learning methods to identify the most optimal model to predict hepatotoxicity and risk factors associated with low-dose MTX.

## 2. Materials and Methods

### 2.1. Study Setting and the Study Population

This retrospective study was conducted at the West China Hospital of Sichuan University, a large tertiary teaching hospital in China. This hospital uses an electronic medical record (EMR) and bar code systems to document medication administrations. The study inclusion criteria were (1) patients with immune system diseases and (2) treated with low-dose MTX (≤30 mg per week) [33] during hospital stays at the West China Hospital of Sichuan University. Patients who were treated with other doses of MTX were excluded. The study period was from 1 January 2018 to 31 December 2019.

### 2.2. Data Extraction

A two-stage review process for medical records was conducted to identify the presence of hepatotoxicity. In the first stage, two trained clinical pharmacists (Hu and Wang) independently reviewed each medical record for the presence of hepatotoxicity. The following sections of the charts were reviewed: basic patient information, diagnostic and progress notes, medication charts, laboratory data, surgical records, nursing flow sheets, and admission and discharge documents. In the second stage, a physician reviewed all medical records identified in the first stage to determine the presence of hepatotoxicity. Disagreements were resolved through a team discussion. Because the clinical interventions usually preceded patients reaching clinical diagnostic criteria of drug-induced liver injury [34], hepatotoxicity was defined as elevated liver enzymes > 1.25 of the upper limits of normal (ULN) and outcomes of liver failure, fibrosis, cirrhosis, or death. The hospital standard cut-off values for ALT are 40 IU/L for women and 50 IU/L for men. AST values are 35 IU/L for women and 40 IU/L for men. Alkaline phosphatase (ALP) values are 135 IU/L for women and 160 IU/L for men. Data collection was carried out from September 2020 to January 2021.

Based on data from included patients’ records, risk factors were screened from multiple patient characteristics to establish a prediction model. Specifically, the following variables were documented: age, gender, height, weight, alcohol use, history of liver diseases (hepatitis B, hepatitis C, and nonalcoholic fatty liver disease), admission, discharge, blood lipid level, antibiotics, other immunosuppressive agents, and Chinese patent medicines. The scores of all risk factors were calculated using the machine learning method and represented by a ranking figure. The factor with a higher score had a more significant impact on the occurrence of hepatotoxicity. Factors with a score of zero were removed because they did not affect the prediction. A Shapley Additive Explanations (SHAP) figure demonstrated the positive or negative correlations between risk factors and hepatotoxicity. Risk factors with a large sample size would affect their impact on the SHAP figure. The SHAP figure was developed by using Python software (version 3.7, Python Software Foundation, Wilmington, DE, USA)

### 2.3. Model Development

Missing data were imputed using the missForest method, and variables with more than 30% missing data were discarded [35]. Patients included were randomly stratified (8:2) into the training set for modeling development and the testing set to evaluate the performance of the models. Using the selected risk factors as covariates, eight machine-learning models were established and analyzed using algorithms including eXtreme Gradient Boosting (XGBoost), AdaBoost, CatBoost, Gradient Boosting Decision Tree (GBDT), Light Gradient Boosting Machine (LightGBM), Tree-based Pipeline Optimization Tool (TPOT), Random Forest (RF), and Artificial Neural Network (ANN). The area under the curve (AUC) of the receiver operating characteristic (ROC) curve, representing the overall ability to classify and predict, is considered the primary metric for evaluating and comparing models. The accuracy, precision, sensitivity, specificity, recall, F1 scores, and average precision (AP) of precision-recall curve were also calculated. These metrics were used to assess the model performance comprehensively. The best-performing model was selected to establish a hepatotoxicity prediction model associated with low-dose MTX. The missForest method and machine learning models were developed and validated with open-source packages in Python software (version 3.7).

### 2.4. Statistical Analysis

Categorical variables were summarized using frequency counts and percentages, and continuous variables were presented as means with standard deviations (SD) or medians with ranges. Comparisons between the training set and the testing set were made using the nonparametric Mann–Whitney U test for continuous variables and the χ^2^ test for categorical variables. By convention, *p* values of less than 0.05 were considered statistically significant. These analyses were performed using the SPSS 25.0 software (IBM Information Management, Chicago, IL, USA).

## 3. Results

### 3.1. Study Population

A total of 2080 medical records were registered in the cohort during the study period, and 782 patients were enrolled in this study. The following patients were excluded: 171 had duplicate records, 588 were on high-dose MTX, and 539 had low-dose MTX as discharge medication (Figure 1). Among the patients enrolled, the mean age was 47.85 ± 15.56 years (range from 10 to 87 years), and the females represented 54.99% (430/782). The average body mass index (BMI) was 22.72 ± 3.91 kg/m^2^ (range from 13.27 to 41.14 kg/m^2^). A total of 279 (35.68%) patients experienced hepatotoxicity. Among these variables analyzed, BMI had 53 missing data points (6.78%) imputed using the missForest method. There was no significant difference between the processed data and the original data. The enrolled patients were divided into training and testing sets in a ratio of 8:2, with 625 and 157 patients, respectively. There were no significant differences in any variables between the training and testing sets (*p* > 0.05) (Table 1).

### 3.2. Model Performance

The visual comparisons of the eight models in the total population are shown in Figure 2, including the precision-recall and the ROC curves. Random Forest achieved the highest AUC of 0.97, followed by XGboost (AUC = 0.94), Catboost (AUC = 0.91), LightGBM (AUC = 0.87), and TPOT (AUC = 0.78). The ROC curves of Adaboost, ANN, and GBDT were low, only 0.69, 0.65, and 0.53, respectively. The precision, accuracy, sensitivity, specificity, recall, and F1 values of the eight models are shown in Table 2.

Adaboost and Random Forest had the highest accuracy (64.33%). Adaboost had the highest precision value (51.35%), followed by GBDT (50.94%). GBDT had the highest sensitivity value (41.07%), followed by Adaboost (33.93%) and XGboost (33.93%). GBDT had the highest specificity value (30.69%), followed by XGboost (24.75%). GBDT had the highest recall value (41.07%), followed by Adboost (33.93%) and XGboost (33.93%). GBDT had the highest F1 value (41.82%), followed by Adaboost (40.86%). These results showed that Adaboost had slight advantages in precision and accuracy with good recall, sensitivity, and F1 values. Adaboost had a significantly lower AUC than Random Forest (0.69 *versus* 0.97). After general consideration of the prediction performance, Random Forest was selected to predict the hepatotoxicity associated with low-dose MTX.

### 3.3. Hepatotoxicity and Risk Factors

A total of 279 patients experienced hepatotoxicity, with an incidence rate of 35.68%. The importance score ranking in the Random Forest model is shown in Figure 3. Importance scores were above zero for all risk factors, indicating that they had a greater or lesser impact on prediction. Among risk factors, the highest score was BMI (0.237), followed by age (0.198), number of drugs (0.151), and number of comorbidities (0.144), demonstrating their importance in hepatotoxicity associated with low-dose MTX.

The SHAP values of the risk factors are shown in Figure 4. When analyzed by the following risk factors (the number of comorbidities, the number of drugs, the use of antibiotics, male gender, the use of alcohol, infectious liver disease, dyslipidemia, and the history of kidney disease). The color of the dot became redder as the SHAP value increased. The color was bluer when the SHAP value decreased. The color changes showed degrees of the positive impact of these factors on the risk of hepatotoxicity. In contrast, risk factors, including BMI and doses of folic acid, showed negative effects. Type 2 diabetes, taking MTX for the first time, other immunosuppressive agents, age, and Chinese patent medicines showed unclear influence.

## 4. Discussion

An effective prediction model is necessary to prevent the hepatotoxicity associated with low-dose MTX. In real-world studies, the variables are not independent but are related nonlinearly. Multivariate analysis methods are challenging for capturing complex relationships. Therefore, we innovatively attempted to apply machine-learning methods that can capture nonlinear relationships between variables. Machine learning can explore risk factors and establish a prediction model for hepatotoxicity associated with low-dose MTX through data learning. Our retrospective study analyzed 15 risk factors for hepatoxicity. The BMI with missing data was imputed using the missForest method, which has been shown to successfully handle missing values, particularly in data sets that include different variables [35]. The results did not show significant differences between the processed and original data.

The eight machine learning methods, including XGBoost, AdaBoost, CatBoost, GBDT, LightGBM, TPOT, RF, and ANN, were applied to establish a prediction model. In these methods, the XGBoost, AdaBoost, CatBoost, GBDT, and LightGBM are boosting algorithms in machine learning. GBDT can combine the predictions from multiple decision trees to generate the final predictions, while it can hardly be adapted to dynamic online data generation, and it tends to be ineffective when facing sparse categorical features [36]. The working procedure of XGBoost is the same as GBDT. XGBoost includes a variety of regularization techniques that can reduce overfitting and improve overall performance, which makes XGBoost slightly better than GBDT. LightGBM is a fast, distributed, high-performance gradient-boosting framework based on a decision tree algorithm. LightGBM uses a histogram-based algorithm, i.e., it buckets continuous feature values into discrete bins that fasten the training procedure [37]. CatBoost is also based on GBDT and has the following two innovations: ordered target statistics and ordered boosting [38]. Therefore, CatBoost works well with the default set of hyperparameters, and the users do not have to spend a lot of time tuning the hyperparameters [38]. Adaboost is relatively robust to overfitting in low-noise datasets. While it is easily defeated by noisy data, the efficiency of the algorithm is highly affected by outliers as the algorithm tries to fit every point perfectly [39]. Random Forest is a bagging algorithm that uses bootstrap aggregation of multiple regression trees to reduce the risk of overfitting and combine the predictions of many trees to produce more accurate predictions [40]. Therefore, Random Forest has a good classification effect for most data. TPOT can automatically optimize feature transformation, feature selection, feature construction, model selection, and parameter optimization via genetic programming using a tree-based structure [41]. The design of ANNs is based on the human brain’s neural network. Neurons in the different layers have their own missions to solve problems, which can be analogous to factory production lines [42]. As a type of parallel distributed system driven by mass data, ANNs are free from the requirements of logical or mathematical associations known beforehand [42].

The performance of different machine learning algorithms should be based on the characteristics of the dataset. Therefore, the choice of models should be based on the calculation results. The results showed that these machine algorithms performed well, especially the Random Forest. The Random Forest showed that its AUC was 0.97. The accuracy and precision were 64.33% and 50.00%, respectively. Both the recall and the F1 scores were satisfactory. Random Forest outperformed other models selected to build the prediction model for hepatotoxicity associated with low-dose MTX.

Analysis of risk factors showed that all 15 variables helped predict low-dose MTX-related hepatotoxicity. The top ten significant risk factors included BMI, age, number of drugs and comorbidities, doses of folic acid, antibiotic use, gender, immunosuppressive agents, taking MTX for the first time, and alcohol use, suggesting physicians should pay more attention to these factors and take the corresponding prevention measures. BMI was considered the most critical risk factor, which had a negative relationship with hepatotoxicity, demonstrating that patients with lower BMI were more likely to experience hepatotoxicity. Therefore, the dose of MTX should be individualized based on height and weight to avoid hepatoxicity. Male gender was also identified as an important risk factor in our study. However, the causal relationship between gender and hepatotoxicity associated with low-dose MTX remains controversial [43,44] and requires further research.

The importance of the number of drugs, the number of comorbidities, and the use of antibiotics was also confirmed. As the primary organ for drug metabolism, the liver is more vulnerable to damage by drugs, active metabolites, or drug interactions [45,46,47]. Multiple drug treatments and comorbid diseases can increase the risks of polypharmacy, drug interactions, and even medication errors [48,49], increasing the risk of hepatotoxicity. Antibiotics are the most common cause of liver damage [50]. However, the potential for liver injury caused by antibacterial drugs was underestimated [51]. Several real-world studies showed that antibiotic-induced liver injury ranged from 13.5% to 65% [52,53,54]. Therefore, to avoid hepatotoxicity during MTX therapy, simplifying treatment regimens should be an important measure for the benefit of patients.

Alcohol consumption is well known to harm the liver, particularly in excess [55]. The American College of Rheumatology and the British Society of Rheumatology recommend limiting alcohol intake for patients on MTX treatment [56,57]. Similarly, we found a positive relationship between alcohol use and hepatotoxicity associated with low-dose MTX. Although the importance score for alcohol consumption was not high in this study due to the relatively small number of patients who drank alcohol, we still recommend limiting or avoiding alcohol intake.

Supplementation with folic or folinic acid during MTX treatment can ameliorate ADEs. Worldwide guidelines currently support the coadministration of folic acid with MTX. The recommended doses range from 0.5 to 2 mg per day [57,58]. However, several studies have suggested that high-dose folinic acid supplementation may reduce the beneficial effects of MTX [59,60,61]. In our study, patients taking high-dose folic acid had a high risk of liver injury. Among patients taking more than 15 mg of folic acid a week, the incidence of liver injury was 41.94%. In contrast, the incidence of liver injury in patients taking 5–10 mg/week was only 34.54%. Furthermore, 59 patients in this study did not take folic acid during MTX treatment, and their liver injury rate was up to 45.76%. Therefore, we recommend daily supplementation with folic acid during MTX treatment.

Metabolic syndrome is a biochemical and clinical condition characterized by visceral obesity, dyslipidemia, hyperglycemia, and hypertension [62,63]. Disorders associated with metabolic syndrome can be significant risk factors for fibrosis and the progression of liver damage. Type 2 diabetes contributed to the biological processes that drove the severity of nonalcoholic fatty liver disease, which was the leading cause of developing chronic liver diseases [64,65]. Several studies showed that nonalcoholic steatohepatitis and hyperlipidemia contributed to MTX hepatotoxicity in patients with psoriasis [43,66]. These were consistent with our results that type 2 diabetes and hyperlipidemia were significant risk factors for hepatotoxicity associated with low-dose MTX.

Hepatitis B and hepatitis C can cause liver damage, increasing the risk of liver toxicity and even liver fibrosis and cirrhosis in patients taking MTX [67]. Infectious liver disease was one of the important risk factors for hepatotoxicity in this study, while its importance score was not high. The reason might be that patients with infectious liver disease were only 4.1% of the study sample. For health and safety reasons in China, many physicians choose other alternative treatments for patients with infectious liver disease instead of MTX. Similarly, only six patients had a history of kidney disease in this study. Therefore, the importance score for the history of kidney disease was low.

Our study has the following limitations (1) knowledge about specific risk factors is still lacking in this study. Although factors such as taking MTX for the first time, other immunosuppressive agents, age, and Chinese patent medicines affected the occurrence of hepatotoxicity, the direction of influence of these factors was unclear. These factors could be influenced by other factors, such as drug regimens (the number of drugs and drug interactions), gender, and BMI; (2) the sample size was small. Future studies should include more patient data from different health care centers; (3) long-term studies are required to verify the association of these risk factors with liver fibrosis or cirrhosis.

## 5. Conclusions

Machine learning can be applied to establish the prediction model for low-dose hepatotoxicity associated with MTX. The model can help to improve medication safety in patients taking methotrexate in clinical practice. However, due to the above limitations, further studies are required to test our findings.

## Figures and Tables

**Figure 1 jcm-12-01599-f001:**
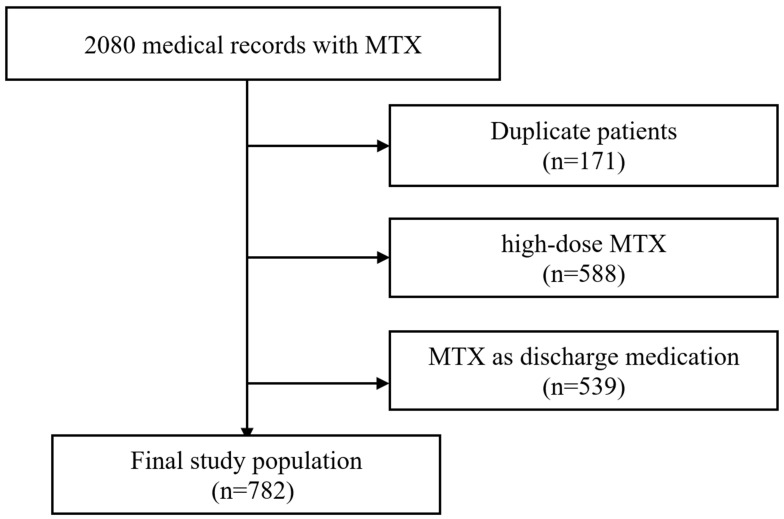
The flow chart illustrating patient selection.

**Figure 2 jcm-12-01599-f002:**
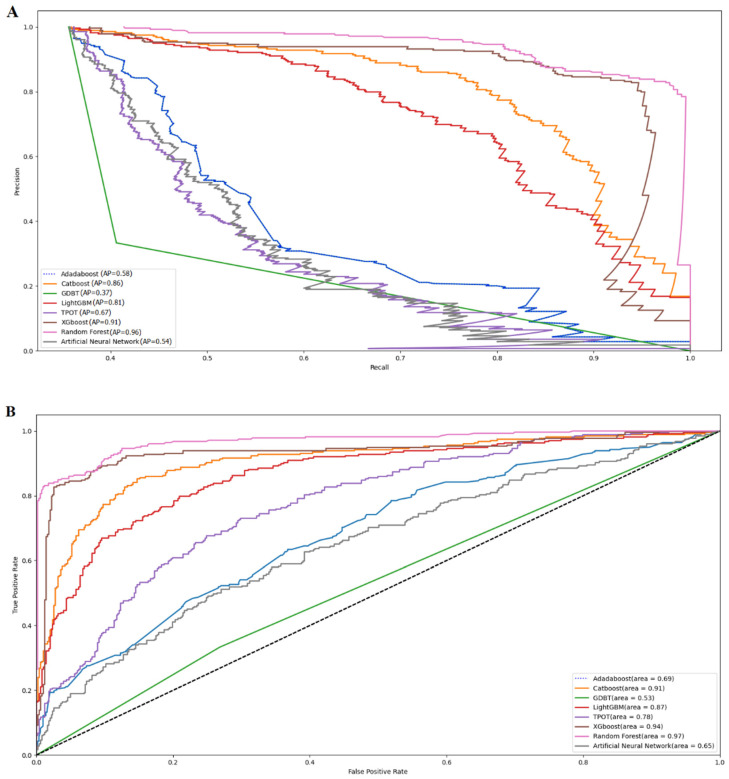
Visual presentation of eight machine learning models (**A**) the precision-recall curve, and (**B**) the receiver operating characteristic (ROC) curve.

**Figure 3 jcm-12-01599-f003:**
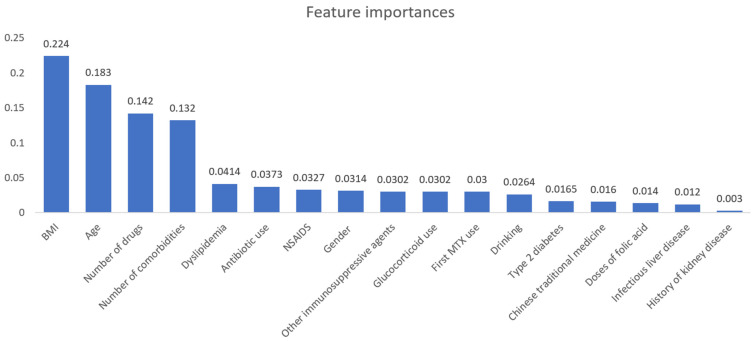
Importance score ranking for risk factors.

**Figure 4 jcm-12-01599-f004:**
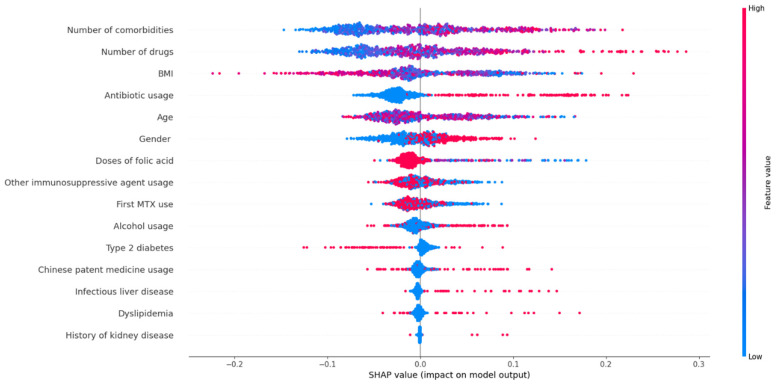
SHAP values of the important risk factors.

**Table 1 jcm-12-01599-t001:** The characteristics of patients.

Variable	Total (*n* = 782)	Training Set (*n* = 625)	Testing Set (*n* = 157)	*p*
Hepatotoxicity				
Yes	279	223	56	1.00
No	503	402	101	
Gender				
Male	352	281	71	1.00
Female	430	344	86	
Age (years)	47.85 ± 15.56 (10–87)	47.67 ± 15.63 (14–78)	48.58 ± 15.33 (10–87)	0.40
First time taking MTX				
Yes	501	400	101	1.00
No	272	225	56	
Body mass index * (kg/m^2^)	Original data			
	729	586	143	
	22.72 ± 3.91 (13.27–41.14)	22.78 ± 3.94 (13.27–41.14)	22.45 ± 3.77 (13.74–37.13)	0.71
	Processed data			
	22.68 ± 3.81 (13.27–41.14)	22.77 ± 3.86 (13.27–41.14)	22.38 ± 3.23 (13.74–37.13)	0.37
Alcohol use				
Yes	164	130	34	0.83
No	618	495	123	
History of kidney disease				
Yes	6	5	1	1.00
No	776	620	156	
History of liver disease				
Yes	32	23	9	0.26
No	750	602	148	
Number of comorbidities	4.98 ± 2.97 (1–17)	4.90 ± 2.90 (1–17)	5.28 ± 3.23 (1–16)	0.69
Type 2 diabetes				
Yes	69	59	10	0.27
No	713	566	147	
Hyperlipidemia				
Yes	41	30	11	0.32
No	741	595	146	
Folate supplementation				
Yes	723	575	148	0.40
No	59	50	9	
Doses of folic acid/week	9.19 ± 3.33 (0–35)	9.16 ± 3.39 (0–35)	9.29 ± 3.07 (0–15)	0.05
NSAIDs use				
Yes	276	231	45	0.06
No	506	394	112	
Glucocorticoid use				
Yes	441	350	91	0.72
No	341	275	66	
Antibiotics use				
Yes	153	117	36	0.26
No	629	508	121	
Other immunosuppressive agent use				
Yes	446	351	95	0.37
No	336	274	62	
Number of medications	5.91 ± 2.93 (0–24)	5.97 ± 2.93 (0–24)	5.67 ± 2.92 (0–18)	0.92
Chinese patent medicines use				
Yes	68	56	12	0.75
No	714	569	145	

*: The difference between processed data and original data was not statistically significant (*p* values were 0.88, 0.94, and 0.87 in the total, training set, and testing set, respectively). NSAID: non-steroidal anti-inflammatory drugs.

**Table 2 jcm-12-01599-t002:** Model performance.

Models	Precision	Accuracy	Sensitivity	Specificity	Recall	F1
LightGBM	40.00%	59.87%	25.00%	20.79%	25.00%	30.77%
GBDT	50.94%	59.24%	41.07%	30.69%	41.07%	41.82%
Adaboost	51.35%	64.33%	33.93%	17.81%	33.93%	40.86%
Catboost	42.86%	60.51%	32.14%	23.76%	32.14%	36.73%
XGboost	43.18%	60.51%	33.93%	24.75%	33.93%	38.00%
Random Forest	50.00%	64.33%	32.14%	17.82%	32.14%	39.13%
TPOT	43.90%	61.15%	32.14%	22.77%	32.14%	37.11%
ANN	36.36%	62.42%	7.14%	6.93%	7.14%	11.94%

## Data Availability

The data presented in this study are available in the present manuscript.

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
