# Peer review of "Predicting Hepatotoxicity Associated with Low-Dose Methotrexate Using Machine Learning"

_jcm, 2023, doi:10.3390/jcm12041599_

Round 1
Reviewer 1 Report
In this manuscript, the authors indicate the effect Predicting hepatotoxicity associated with low-dose methotrexate using machine learning.
1. After sentences ‘These risk factors include alcohol use, history of liver disease, obesity, type 2 diabetes, history of significant exposure to hepatotoxic drugs or chemicals, lack of folate supple-mentation, and hyperlipidemia’’ It should be given information. For this purpose the authors can look at the following articles for introduction section: Journal of Biomolecular Structure and Dynamics 40 (1), 77-85 and Open Chemistry 19 (1), 347-357.
2. In discussion section, After sentences ‘As the primary organ for drug metabolism, the liver is more vulnerable to damage by drugs, active metabolites, or drug interactions. It should be given information. For this purpose the authors can look at the following articles for introduction section: Applied biochemistry and biotechnology 190 (2), 437-447, Comparative biochemistry and physiology part C: toxicology & pharmacology and ChemistrySelect 6 (43), 11915-11924.
3. In discussion section, After sentences ‘Metabolic syndrome is a biochemical and clinical condition characterized by visceral obesity, dyslipidemia, hyperglycemia, and hypertension. It should be given information. For this purpose the authors can look at the following articles for introduction section: Archives of physiology and biochemistry 125 (5), 387-395 and Journal of Sleep Research 29 (2), e12956.
4. In discussion section, After sentences ‘’ Type 2 diabetes contributed to the biological processes that drove the severity of nonalcoholic fatty liver disease, which was the leading cause of developing chronic liver’’ It should be given information. For this purpose the authors can look at the following articles for introduction section Bioorganic Chemistry 102, 104110 and Journal of Molecular Structure 1224, 129446.
5. The image quality of the figures can be improved a little more.
6. Grammaticals errors found in the manuscript. It should be corrected.
Author Response
- After sentences ‘These risk factors include alcohol use, history of liver disease, obesity, type 2 diabetes, history of significant exposure to hepatotoxic drugs or chemicals, lack of folate supple-mentation, and hyperlipidemia’’ It should be given information. For this purpose the authors can look at the following articles for introduction section: Journal of Biomolecular Structure and Dynamics 40 (1), 77-85 and Open Chemistry 19 (1), 347-357.
I am so sorry. I have read the introduction section of Journal of Biomolecular Structure and Dynamics 40 (1), 77-85, while it may be not relevant to the content of this manuscripts.
I do not find Open Chemistry 19 (1), 347-357 by using PubMed or Google Scholar. Could you provide the PMID?
- In discussion section, After sentences ‘As the primary organ for drug metabolism, the liver is more vulnerable to damage by drugs, active metabolites, or drug interactions. It should be given information. For this purpose the authors can look at the following articles for introduction section: Applied biochemistry and biotechnology 190 (2), 437-447, Comparative biochemistry and physiology part C: toxicology & pharmacology and ChemistrySelect 6 (43), 11915-11924.
I have add Applied biochemistry and biotechnology 190 (2), 437-447, Comparative biochemistry to references.
I do not find physiology part C: toxicology & pharmacology and ChemistrySelect 6 (43), 11915-11924 by using PubMed or Google Scholar. Could you provide the PMID?
- In discussion section, After sentences ‘Metabolic syndrome is a biochemical and clinical condition characterized by visceral obesity, dyslipidemia, hyperglycemia, and hypertension. It should be given information. For this purpose the authors can look at the following articles for introduction section: Archives of physiology and biochemistry 125 (5), 387-395 and Journal of Sleep Research 29 (2), e12956.
I have add Archives of physiology and biochemistry 125 (5), 387-395 to references.
I am so sorry. I have read the introduction section of Journal of Sleep Research 29 (2), e12956, while it may be not relevant to the content of this manuscripts.
- In discussion section, After sentences ‘’ Type 2 diabetes contributed to the biological processes that drove the severity of nonalcoholic fatty liver disease, which was the leading cause of developing chronic liver’’ It should be given information. For this purpose the authors can look at the following articles for introduction section Bioorganic Chemistry 102, 104110 and Journal of Molecular Structure 1224, 129446.
I have add Bioorganic Chemistry 102, 104110 to references.
I do not find Journal of Molecular Structure 1224, 129446 by using PubMed or Google Scholar. Could you provide the PMID?
- The image quality of the figures can be improved a little more.
I have updated the figure1 and 3.
The Figure 2 and 4 were come from Python, so the quality of them could not be updated
- Grammaticals errors found in the manuscript. It should be corrected.
I have corrected the Grammaticals errors.If the grammar still does not meet your requirements, I will seek professional help

Reviewer 2 Report
In this study, Hu et al. seek the efficacy of machine learning to predict MTX-induced hepatotoxicity. The topic is unique and tempting. I have a few minor comments for this manuscript. In Figure 2, results look significantly different in different machine learning. Statistical analysis to calculate P values should be performed to see if the difference is significant between algorithms. Random Forest looks best, but XGboost is close, so if there is no statistical difference between these two, the authors should not describe as if Random Forest is the best. In addition, the authors should describe or summarize characteristics of each algorithm and speculate the reason for the difference. For example, GDBT shows the worst, but does this mean that GDBT is poorly designed or not powerful or not suitable for this type of analysis or it requires more parameters for accurate prediction or what? The authors mention that this study is limited and some parameters are missing. With Figure 2 data, the readers may think that GDBT is useless, but I think that is misleading. The authors should discuss the limitations of current machine learning and algorithms, pros and cons, useful situations and not appropriate situations, strength and weak points for each algorithms, or something like that.
Author Response
In this study, Hu et al. seek the efficacy of machine learning to predict MTX-induced hepatotoxicity. The topic is unique and tempting. I have a few minor comments for this manuscript. In Figure 2, results look significantly different in different machine learning. Statistical analysis to calculate P values should be performed to see if the difference is significant between algorithms. Random Forest looks best, but XGboost is close, so if there is no statistical difference between these two, the authors should not describe as if Random Forest is the best.
Fieger 2 included the precision–recall curve and the receiver operating characteristic (ROC) curve.
The results in the lower left and right corner of the precision–recall curve and receiver operating characteristic (ROC) curve can show the different of these models. Random Forest looks best, but XGboost is close, so if there is no statistical difference between these two, the authors should not describe as if Random Forest is the best. These differences cannot be shown by P values.
In the precision–recall curve, the AP of RF is 0.96, following XGboost (0.91). And in receiver operating characteristic (ROC) curve, the AREA of RF is 0.97, following XGboost (0.94). These results can show the RF is superior to XGboost. In Table 2, the Precision, Accuracy, Specificity, and F1 of RF are all better than XGboost. So we considered the RF is superior to XGboost.
In addition, the authors should describe or summarize characteristics of each algorithm and speculate the reason for the difference. For example, GDBT shows the worst, but does this mean that GDBT is poorly designed or not powerful or not suitable for this type of analysis or it requires more parameters for accurate prediction or what? The authors mention that this study is limited and some parameters are missing. With Figure 2 data, the readers may think that GDBT is useless, but I think that is misleading. The authors should discuss the limitations of current machine learning and algorithms, pros and cons, useful situations and not appropriate situations, strength and weak points for each algorithms, or something like that.
I will analyze this in the discussion section

Round 2
Reviewer 1 Report
The manuscript can be accepted this form.